# Modelling microplastic bioaccumulation and biomagnification potential in the Galápagos penguin ecosystem using Ecopath and Ecosim (EwE) with Ecotracer

Karly McMullen[1]*, Félix Hernán Vargas[2], Paola Calle[3], Omar Alavarado-Cadena[3], Evgeny A. Pakhomov[1,4], Juan José Alava[1]*

**1** Ocean Pollution Research Unit, Institute for the Oceans and Fisheries, University of British Columbia, Vancouver, British Columbia, Canada, **2** The Peregrine Fund, Isla Santa Cruz, Galápagos Islands, Ecuador, **3** Escuela Superior Politécnica del Litoral, ESPOL, Facultad de Ciencias de la Vida, ESPOL Polytechnic University, Guayaquil, Ecuador, **4** Department of Earth, Ocean and Atmospheric Sciences, University of British Columbia, Vancouver, British Columbia, Canada

* k.mcmullen@oceans.ubc.ca (KM); j.alava@oceans.ubc.ca (JJA)

**Data Availability Statement:** The data underlying some results presented in the study are available from D. Ruiz and M. Wolff (2011): https://www.

## Abstract

Bioaccumulation and biomagnification of anthropogenic particles are crucial factors in assessing microplastic impacts to marine ecosystems. Microplastic pollution poses a significant threat to iconic and often endangered species but examining their tissues and gut contents for contaminant analysis via lethal sampling is challenging due to ethical concerns and animal care restrictions. Incorporating empirical data from prey items and fecal matter into models can help trace microplastic movement through food webs. In this study, the Galápagos penguin food web served as an indicator species to assess microplastic bioaccumulation and biomagnification potential using trophodynamic Ecopath with Ecosim (EwE) modelling with Ecotracer. Empirical data collected from surface seawater near Galápagos penguin colonies, zooplankton, penguin prey, and penguin scat in October 2021 were used to inform the ecosystem model. Multiple scenarios, including a 99% elimination rate, were employed to assess model sensitivity. Model predictions revealed that microplastics can bioaccumulate in all predator-prey relationships, but biomagnification is highly dependent on the elimination rate. It establishes the need for more research into elimination rates of different plastics, which is a critical missing gap in current microplastic ecotoxicological and bioaccumulation science. Compared to empirical data, modelling efforts underpredicted microplastic concentrations in zooplankton and over-predicted concentrations in fish. Ultimately, the ecosystem modelling provides novel insights into potential microplastics' bioaccumulation and biomagnification risks. These findings can support regional marine plastic pollution management efforts to conserve native and endemic species of the Galápagos Islands and the Galápagos Marine Reserve.

sciencedirect.com/science/article/pii/ S138511011100075X Model data can be accessed on SEANOE (McMullen K, Hernán VF, Calle P, Alavarado-Cadena O, Pakhomov E, Alava JJ. Modelling microplastic bioaccumulation and biomagnification potential in the Galápagos penguin ecosystem using Ecopath and Ecosim (EwE) with Ecotracer. SEANOE. 2023. https://doi. org/10.17882/97201).

**Funding:** This research was partially funded by the Nippon Foundation via the Nippon Foundation Marine Litter Project at the Institute for the Oceans and Fisheries, University of British Columbia, awarded to J. J. Alava and E. A. Pakhomov under award number F19-02677 (NIPPFOUN 2019) in collaboration with and via the Nippon Foundation-Ocean Nexus Center (Dr. Yoshitaka Ota) at University of Washington [https://oceannexus.uw. edu/about/]. This research was partially supported by the NSERC Discovery Grant RGPIN-2014-05107 to E. A. Pakhomov [https://www. nserccrsng.gc.ca/professors-professeurs/grants-subs/dgigp-psigp_eng.asp]. Both the Nippon Foundation and NSERC funders had no involvement in the study design, data collection and analysis, decision to publish, or preparation of the manuscript.

**Competing interests:** The authors have declared that no competing interests exist.

## 1. Introduction

In the Plasticene era, escalating plastic production has outpaced plastic elimination efforts, leading to its surge in the environment [1, 2]. Over 170 trillion plastic particles now float in the global ocean, with this estimate consistently rising [3–5]. These particles, many of which are considered microplastics (plastic particles 1000 nm to < 5 mm in size), become bioavailable to marine organisms, and have been widely documented in fish [6, 7] and invertebrates [8], including zooplankton [9], and, to a lesser extent, seabirds [10–13], marine mammals [14–17], and sea turtles [14, 18–20]. Microplastics, due to their size and pervasive nature, enter biota through various pathways, including direct intake via water [21–24] and air [21, 25], as well as indirectly through contaminated prey [16, 21, 26–29]. The latter may represent a major pathway of microplastic ingestion for top predators [16]. These particles can cause physical damage to the gut and digestive tract, endocrine disruption due to sorption and leaching of toxic chemicals, and can impair feeding ability and predator avoidance [30–34]. These sublethal but chronic health impacts may have cascading effects at the species or ecosystem level [35].

Bioaccumulation, bioconcentration, and biomagnification are common toxicokinetic and bioaccumulation science concepts often used in ecotoxicological and environmental risk assessments for water soluble (hydrophilic) chemicals such as pesticides, metals, pharmaceuticals, and personal care products [36], hydrocarbons, and ionic or neutral chemicals such as per- and polyfluoroalkyl substances (PFAS), as well as for hydrophobic or lipophilic substances, including persistent organic pollutants (POPs) [37–39]. Bioaccumulation refers to the increase of a pollutant, i.e. microplastics, in an organism over time, or the gradual net uptake from all environmental compartments, including the surrounding environment and the food items [21, 28, 29, 39, 40]. Bioconcentration is a subcategory of bioaccumulation and refers to the gradual buildup of a contaminant in an organism from the water alone, excluding uptake from prey [29, 37]. Conversely, biomagnification implies an increase of the contaminant concentration at each trophic level of the food web. The contaminant concentration amplifies through the food web and organisms at higher trophic levels or apex predators exhibit elevated contaminant concentrations compared to organisms at lower trophic levels [39, 41].

These concepts have recently been applied to microplastic science [21, 26, 28, 29, 36, 42]. A recent literature review across multiple empirical studies shows evidence for bioaccumulation of microplastics within trophic levels [42]. Studies have also established trophic transfer of microplastics in natural-like laboratory settings [16, 27]. However, while species-specific bioaccumulation is likely to occur in marine species and it is a function of the elimination rate [21], evidence for biomagnification of microplastics between trophic levels is lacking [29, 42, 43]. The bioaccumulation and biomagnification capacity of microplastics are a cause for concern due to potential impacts on high trophic level species, and therefore require further research [40, 42].

Ecosystem and bioaccumulation modelling in tandem with the application of several ecotoxicological and bioaccumulation metrics can help to determine exposure levels and bioaccumulation potential, without needing to conduct invasive and lethal sampling or dissection of live individual organisms of threatened, endemic marine species. Within this rationale, food web bioaccumulation modelling is a feasible and useful tool to predict and estimate the bioaccumulation potential and ecotoxicological risks of microplastics to higher trophic level species [21].

The Galápagos Islands offer a unique opportunity to study microplastic bioaccumulation and biomagnification in relatively isolated and simplistic food webs. Recent studies have highlighted presence of microplastics in the archipelago [44, 45], but little is known about the trophic transfer and accumulation of microplastics in endemic and endangered marine species

in the unique food webs of the islands. Galápagos seabirds, including the Galápagos penguin (*Spheniscus mendiculus*), may serve as sentinel species for biomonitoring microplastic pollution [46]. The Galápagos penguin is the only tropical, non-migrating penguin species with a decreasing population currently estimated at 1200 individuals [47]. This species faces several challenges from intense El Niño events and ocean warming [48, 49] to ocean plastics [44].

Using the Galápagos penguin as "the canary in the coal mine," this study predicted the bioaccumulation and biomagnification of microplastics in their food web. The objectives of this study were to: first, understand the bioaccumulation behaviour of microplastics in the food web of the Galápagos penguin; second, predict the biomagnification potential of microplastics in the food web of the Galápagos penguin; and third, compare these findings to empirical data gathered in the Galápagos. Trophodynamic ecosystem modelling, including basic and advanced Ecopath and Ecosim (EwE) models along with the Ecotracer routine [50], were applied. Empirical seawater and biotic data gathered in the Galápagos were used as uptake and environmental input data and later were compared against the predicted data to assess model bias and corroborate the modelling performance. The research was designed to provide essential insights into the bioaccumulation potential and ecotoxicological risks of microplastics to higher trophic level species and inform regional policies to combat marine plastic pollution in the Galápagos Islands.

## 2. Methods

### 2.1 Ecosystem modelling theory

Ecopath with Ecosim (EwE) is a trophodynamic ecosystem modelling tool used to understand how different species in an ecosystem interact with one another, how they respond to changes in the environment, and how human activities might impact the ecosystem [50]. Ecopath is central to the software suite and provides a mass-balance snapshot of the ecosystem in question, while Ecosim offers a dynamic approach for temporal simulations based on predator-prey interactions using Lotka-Volterra and foraging arena theories [50–52]. The EwE model together integrates biotic and abiotic components in an ecosystem by incorporating the principles of mass balance as well as a set of linear equations that describe and track the average flow of mass and energy between functional groups according to a diet composition matrix. Functional groups can be species or groups of species that have similar life-history or characteristics which are combined into biomass pools. The diet composition matrix is used to represent the flow of mass and energy within the ecosystem. It also accounts for energy losses over time through processes such as respiration, emigration, and decomposition [50–52]. The core principles and mathematical equations of EwE are described in the user guide, which is accessible at http://www.ecopath.org [53].

### 2.2 Model descriptions

Two models were used to simulate microplastic movements through the Galápagos penguin food web, namely, a novel and simplistic food web model for the Galápagos penguin based on the species diet; and a trophic model of the Bolivar Channel Ecosystem (BCE), based on Ruiz and Wolff model [54] which includes an advanced snapshot of the BCE, with functional groups including seabirds, a proportion of which are Galápagos penguins (i.e., ~26% of the functional group 'pool'). The latter incorporates a wider variety of groups and energy flows for a holistic ecosystem view, while the former isolates the Galápagos penguin food web to closely track microplastic movements within the specific food web. Both models were run with the Ecotracer routine to assess and compare bioaccumulation and biomagnification.

**Table 1. Diet composition of Galápagos penguin (GP) EwE model.**

|  | Prey/predator | TL | 1 | 2 | 3 | 4 | 5 | 6 | 7 |
|---|---|---|---|---|---|---|---|---|---|
| 1 | Galápagos Penguin | 3.7 | | | | | | | |
| 2 | Barracuda | 3.6 | 0.05 | | | | | | |
| 3 | Mullet | 2.3 | 0.05 | | | | | | |
| 4 | Anchovy, Herring, Sardines, Salema | 2.7 | 0.81 | 0.6 | | | | | |
| 5 | Decapods | 2.0 | 0.05 | | | | | | |
| 6 | Predatory zooplankton | 2.6 | 0.02 | 0.1 | 0.1 | 0.25 | 0.01 | 0.2 | |
| 7 | Herbivorous zooplankton | 2 | 0.02 | 0.1 | 0.05 | 0.2 | 0.01 | 0.25 | |
| 8 | Macroalgae | 1 | | | 0.05 | | 0.3 | | 0.05 |
| 9 | Microalgae/phytoplankton | 1 | | | 0.2 | 0.2 | 0.2 | 0.2 | 0.6 |
| 10 | Detritus | 1 | | | 0.4 | 0.15 | 0.2 | 0.15 | 0.15 |
| | **Import** | | **0** | **0.1** | **0.2** | **0.2** | **0.28** | **0.2** | **0.2** |
| | **Sum** | | **1** | **1** | **1** | **1** | **1** | **1** | **1** |

Prey and diet composition matrix in the food web of the Galápagos penguin (GP) EwE model, based on the best available data and information from the existing literature and expert knowledge. The numbers preceding the prey/predator name indicate the functional group number. These numbers can be found in the column headers of the primary row, where each number corresponds to the respective functional group. Import indicates the proportion of diet a predator consumes that is not considered within the context of this ecosystem model.

**2.2.1 The Galápagos Penguin (GP) food web model.** The Galápagos Penguin (GP) food web model was constructed using EwE 6.6.8 [53]. The model was developed using a top-down approach by first analyzing the diet of the Galápagos penguin (Table 1). Based on limited available dietary data, the Galápagos penguin is suggested to feed primarily on small planktivorous fish, including sardines (*Sardinops sagax*), piquitingas (*Lile stolifera*), and salema (*Xenocys jessiae*), as well as fry or juveniles of mullets (*Mugil spp.*) [48, 55–57; F. H. Vargas, pers. comm., 16 August 2023]. Closely related to the Galápagos penguin, the Humboldt Penguin (*Spheniscus humboldti*) also feeds on anchovies (*Engraulis ringens*), Araucanian herring (*Strangomera bentincki*), and silverside (*Odontesthes regia*), and to a lesser extent, cephalopods (Patagonian squid, *Doryteuthis gahi* and Humboldt squid, *Dosidiscus gigas*) and crustaceans (stomatopods and isopods) [58]. The diet composition of the Magellanic penguin (*Spheniscus magellanicus*), which is also closely related to the Galápagos penguin, was found to be similar [59]. Assessed penguins primarily ate anchovy (*Engraulis anchoita*) and thornfish (*Bovichtus argentinus*) [59]. This information provided a foundation for creating the GP model based on plausible diet matrix assumptions.

The finalized diet matrix was compared and adjusted based on an existing EwE model available on Ecobase (http://ecobase.ecopath.org/) [60, 61], the Floreana island rocky reef ecosystem model, which includes the Galápagos penguin in the seabird functional group along with other marine species [62], as well as the BCE model [54]. Initially, cephalopods were part of the GP model; however, this species functional group was eventually excluded due to the unavailability of sufficient biomass data and the fact that they were not accounted for in the Floreana and BCE models. Conversely, barracuda (*Barracuda pelicano*) were added to the GP model because they were recognized as seabird prey in the Floreana and BCE models [54, 62]. It is reasonable to conclude that juvenile barracuda are likely to be preyed upon, given the preferred prey size for penguins [58, 63].

The final diet matrix consisted of ten functional groups, as shown in Table 1. Species selection and identification were done using scuba diving field guides [64, 65] as well as using the BCE model [54]. Diet information for penguin prey and lower trophic level species were

**Table 2. List of species and functional groups of the Galápagos penguin (GP) EwE model.**

| Common name | Species name | Source |
|---|---|---|
| Galápagos Penguin | *Spheniscus mendiculus* | NA |
| Barracudas | *Sphyraena idiates* | [64] |
| Mullet | *Mugil galapagensis\*, cephalus, curema* | [54, 64, 65] |
| Anchovy | *Engraulidae: Anchoa naso* | [64] |
| Sardines | *Sardinops sagax sagax* | [64] |
| Herrings | *Clupeidae: Opisthonema berlangai* | [64] |
| Salema | *Xenichthys agassizi, Xenocys jessiae\** | [54, 64, 65] |
| Decapods | *Panulirus gracilis, panulirus penicillatus, Scyllariidea astori\** | [54, 64] |
| Predatory Zooplankton | Functional group *spp.* | [54] |
| Herbivorous Zooplankton | Functional group *spp.* | [54] |

*The asterisk represents the main species referenced to determine biological parameters.

Data is based on the existing literature sources.

accessed through FishBase (https://www.fishbase.se/) and SeaLifeBase (https://www.sealifebase.ca/) [66–68]. Where data was not available, the diet information of the closest related species was used. Functional group biomasses (t/km$^2$) were calculated based on existing models and reasonable estimations of biomass in g/m$^2$. Production per biomass (P/B) was calculated based on the average lifespan of the species (i.e., the inverse of mortality). Galápagos penguins, for example, live up to 15 to 20 years [69], therefore a P/B of 0.07 is reasonable when assuming a mortality of $\frac{1}{15}$ y$^{-1}$. Consumption per biomass (Q/B) was obtained from FishBase and SeaLifeBase [66–68]. The Q/B parameter from FishBase was adjusted and recalculated for barracuda and mullets, given the penguins' preference for smaller-sized prey; therefore, assuming the barracuda and mullet functional groups are comprised of juveniles, the size i.e., max total length (TL), was set to 30 cm for barracuda (*S. idiates*) and 25 cm for mullet (*M. galapagensis*), then FishBase calculations were rerun. Ecotrophic efficiency was left to be calculated by the EwE model. The sources of information and input data can be found in Table 2 & S1 Table in S1 File [48, 54, 57, 62, 70–73].

After inputting all parameter estimates, the model was found to be unbalanced, therefore parameters were manually adjusted until balance was achieved. The Q/B was reduced, and the P/B was increased for fish so that the P/Q ratio was near 0.2–0.3 based on EwE best practices [74, 75]. The biomass of herbivorous and predatory zooplankton was increased to account for their predation. Given the intentionally limited scope of the model and specific focus on the Galápagos penguin, diet import was reasonably assumed for most functional groups given that not all prey were included (e.g., not all barracuda prey).

**2.2.2 Bolivar Channel Ecosystem (BCE) model.** The Bolivar Channel Ecosystem (BCE) model is a more advanced EwE model consisting of thirty functional groups known to inhabit the highly productive, upwelling zone between Isabela and Fernandina islands [54]. The original model was provided by courtesy of the authors to support the assessment of microplastic bioaccumulation in the area. Ruiz and Wolff [54] estimated biomass from observed sightings and censuses where applicable. The model includes one functional group entitled 'seabirds', comprising 74% of flightless cormorants (*Phalacrocorax harrisi*) and 26% Galápagos penguins (*S. mendiculus*). Input parameters underwent a series of resampling through the Ecoranger resampling routine to select a random set of input values from normal distributions of the input parameters. The model is fully described in Ruiz and Wolff [54].

## 2.3 Modelling bioaccumulation of microplastics with EwE Ecotracer routine

The Ecotracer routine is an EwE module used to track and assess the bioaccumulation potential of pollutants in marine food webs over time [50, 76, 77]. Ecotracer uses Ecosim temporal simulations to predict the flow of contaminants through the food web [76–78]. The contaminant concentration over time in each functional group is based on the flow rates from Ecosim, as well as decay, elimination, and physical exchange rates. The linear dynamical equation for changes in contaminant concentration over time for a given functional group (pool) or species and the dynamic changes in contaminant concentration can be found in *Ecopath with Ecosim: A User's Guide Ecosystem effects of invertebrate fisheries View project Ecosystem Fmsy estimations for data rich stocks* [52].

The contaminant concentrations in immigrating and emigrating organisms were considered to be negligible for the purposes of this modelling work to simplify the model and because the Galápagos penguins are endemic and residents of the Galápagos Islands and do not undertake migration.

Input parameters were gathered from observed data from the field [79], laboratory studies, and existing modelling work outlined in Table 3. Microplastic concentrations (particle size > 10 μm) in the environment were obtained [79] and were entered as input data for the initial and inflow environmental concentrations of microplastics in the Ecotracer module for both EwE models. The direct uptake rates were derived from the observed zooplankton ingestion rates of anthropogenic particles [79]. This information was utilized as the exclusive source for the microplastic uptake, based on the premise that zooplankton occupy a pivotal position at the bottom of aquatic food webs and therefore represent the initial point of entry for the bioaccumulation of microplastics in the ecosystem [21]. Considering the purpose of the modelling work, which aimed to evaluate the levels and magnitude of bioaccumulation and biomagnification of microplastics in the Galápagos penguin food web, alternative sources of microplastic exposure such as inhalation of airborne particles or direct uptake from the environment were assumed to be negligible and thus not considered.

Microplastic elimination or egestion rates were adopted from the food web bioaccumulation model developed by Alava [21] or collected from the best available data (See Table 4). Briefly, the egestion or elimination rate of microplastic is computed from the retention time ($\tau$) as the elimination or egestion rate is inversely related to the retention or residency time: $\tau = 1/k_E$; thus, solving for $k_E$:$k_E = 1/\tau$ [21]. Finally, an average decay rate was set at ~0.0283 per year (2.825% per year), based on plastic particles decay estimates (ranging 0.65% to 5% per year), accounting for weight loss due to solar radiation and oxygenation, documented by Everaert et al. [80]. Ecotracer was run for a simulation period of 100 years.

## 2.4 Model scenarios

To explore the microplastic bioaccumulation potential and biomagnification capacity in the food web, the model was run using four different scenarios: (1) A baseline scenario using

**Table 3. Ecotracer environmental parameters, input data, and respective sources used in the Galápagos Penguin (GP) EwE model.**

| Ecotracer Parameter | Environmental Input Data | Source/Details |
|---|---|---|
| Initial concentration (t/km$^2$) | 4.11x10$^{-2}$ | 7.57x10$^{-6}$ g/MP [21]<br>4.11x10$^{-6}$ ± 3.72x10$^{-6}$ average MP g/L [79]<br>Converted to t/km$^2$ based on [81] |
| Base inflow rate (t/km$^2$/y) | 4.11x10$^{-2}$ | Consistent state scenario based on [82] |
| Decay Rate (/year) | 0.0283 | [80] |
| Direct absorption rate (g/kg) | 7.38x10$^{-2}$ | Based on observed zooplankton ingestion rate [79] |

**Table 4. Data values of microplastic retention times to calculate the egestion (elimination) rates with reference sources used for the Galápagos Penguin (GP) EwE model.**

| Group | Conservative Elimination Rate (Short Retention time) | Elimination Rate Based on Literature | Source |
|---|---|---|---|
| Galápagos penguin | <24 hours | 70 hours | [83, 84] |
| Barracuda | <24 hours | 49 days | [85, 86] |
| Mullet | <24 hours | 30 days | [7] |
| Anchovy, Herring, Sardines, Salema | <24 hours | 49 days | [85–87] |
| Decapods | <24 hours | 14 days | [88] |
| Predatory zooplankton | <24 hours | 7 days | [31] |
| Herbivorous zooplankton | <24 hours | 7 days | [31] |

observed environmental anthropogenic particle concentrations [79] and egestion rates from literature; (2) a high environmental concentration scenario, which assumed a higher abundance of microplastics in the environment, based on the upper limit of the standard deviation of the aforementioned observed microplastic abundance data; (3) a low environmental concentration scenario, which assumed a lower abundance of microplastics in the environment, based on the lower limit of the standard deviation of the microplastic abundance data; and finally, (4) a 99% egestion rate scenario which assumes all microplastics are excreted in under 24 hours, using the environmental concentration data from baseline scenario (S2 Table in S1 File). See S1-S4 Tables in S1 File for additional Ecotracer input data.

## 2.5 Bioaccumulation and biomagnification metrics

The application of bioaccumulation and biomagnification criteria and metrics for microplastics were based on Alava [21, 89, 93].

*Bioaccumulation factor (BAF).* The bioaccumulation factor (BAF) was calculated by comparing the amount of microplastics accumulated by aquatic biota to the total concentration of microplastics present in both the aquatic environment and diet (e.g., the predator's prey). A BAF greater than 1 indicates plausible bioaccumulation under steady state (e.g., BAF >1).

$$BAF = \frac{Ci}{Co + Cj} \tag{1}$$

Where $Ci$ is the microplastic concentration in the predator (in units of g/kg), $Co$ is the concentration in the environment and $Cj$ is the concentration in the diet (prey) in units of g/kg.

*Bioconcentration factor (BCF).* The bioconcentration factor (BCF) was calculated by comparing the amount of microplastics accumulated by the aquatic organism or biota (in units of g/kg) to the total concentration of microplastics present in the water or aquatic environment (in units of g/L). A BCF greater than 1 indicates bioconcentration in the organism (e.g., BCF >1).

$$BCF = \frac{Ci}{Cw} \tag{2}$$

Where $Ci$ is the microplastic concentration in biota (g/kg) and $Cw$ is the concentration in the water (g/L).

*Predator-prey biomagnification factor (BMF$_{TL}$).* The predator-prey biomagnification factor (BMF$_{TL}$) was calculated by comparing the amount of microplastics accumulated by the predator in units of g/kg to the total concentration of microplastics (g/kg) in the prey, divided by the difference in trophic levels. A BMF greater than 1 indicates plausible trophic biomagnification

(e.g., $BMF_{TL} > 1$).

$$BMF = \frac{Ci/Cj}{TLi - TLj} \qquad (3)$$

Where $Ci$ is the microplastic concentration in the predator, $Cj$ is the concentration in prey, $TL$ is the trophic level for the predator ($i$) and prey ($j$).

*Trophic magnification factor (TMF).* The trophic magnification factor (TMF) is a commonly employed metric in food web analysis that measures the biomagnification of pollutants at various trophic levels [21, 90–92]. The TMF is calculated as the antilog of the regression slope of the linear regression between the logarithmic-transformed concentrations of microplastics (Log MPs) predicted in the GI tract of organisms of the food web and their trophic level, TL [21]. The linear equation is as follows.

$$Log_{10}MP = a + bTL \qquad (4)$$

Where $b$ is the slope, $TL$ is the trophic level, and $a$ is the y-intercept. The *TMF* can be expressed as the equivalent exponential mathematical terms expressed as TMF = $10^b$, where $b$ is the slope.

$$TMF = 10^b \qquad (5)$$

The TMF (slope, $b$) is statistically evaluated using a significance level ($\alpha$) of 0.05. A TMF > 1 ($b > 0$) indicates that the contaminant biomagnifies in the food web. A TMF < 1 ($b < 0$) indicates trophic dilution of the contaminants, while a TMF = 1 ($b = 0$) indicates no change in contaminant concentrations among organisms of a food web [21, 90].

## 2.6 Sensitivity assessment and model bias

The sensitivity of the model was assessed by testing changes in the environmental concentrations and the functional group elimination rates. This was conducted by comparing the outcomes of the model through four of the different scenarios (S2 Table in S1 File), including high and low environmental concentrations as well as high (e.g., > 24 hours) and low (e.g., < 24 hours) retention times, equivalent to slow (high retention) or fast (low retention) elimination rates, respectively. The most sensitive parameters were determined by assessing the variance in output data according to changes in environmental concentrations and elimination rate.

## 2.7 Model bias

A model bias (*MB*) approach was applied to assess the performance of the food web model and corroborate the projections of microplastics under the three abiotic concentrations' scenarios (scenario 1–3) and one conservative egestion rate scenario (scenario 4). The performance of the model was analyzed in terms of the model bias ratio:

$$MB = \frac{C_{BP,iMP}}{C_{BO,iMP}} \qquad (6)$$

where $C_{BP,iMP}$ and $C_{BO,iMP}$ are the model calculated (predicted) and observed microplastic concentrations in species $i$, respectively. This analysis was done by comparing the projected microplastics concentrations in biota (zooplankton, anchovy, and mullet) yielded by the model simulations to the empirical data measured in free-ranging zooplankton, wild-caught anchovy, and mullets in waters of the Galápagos National Park [79]. The microplastic and

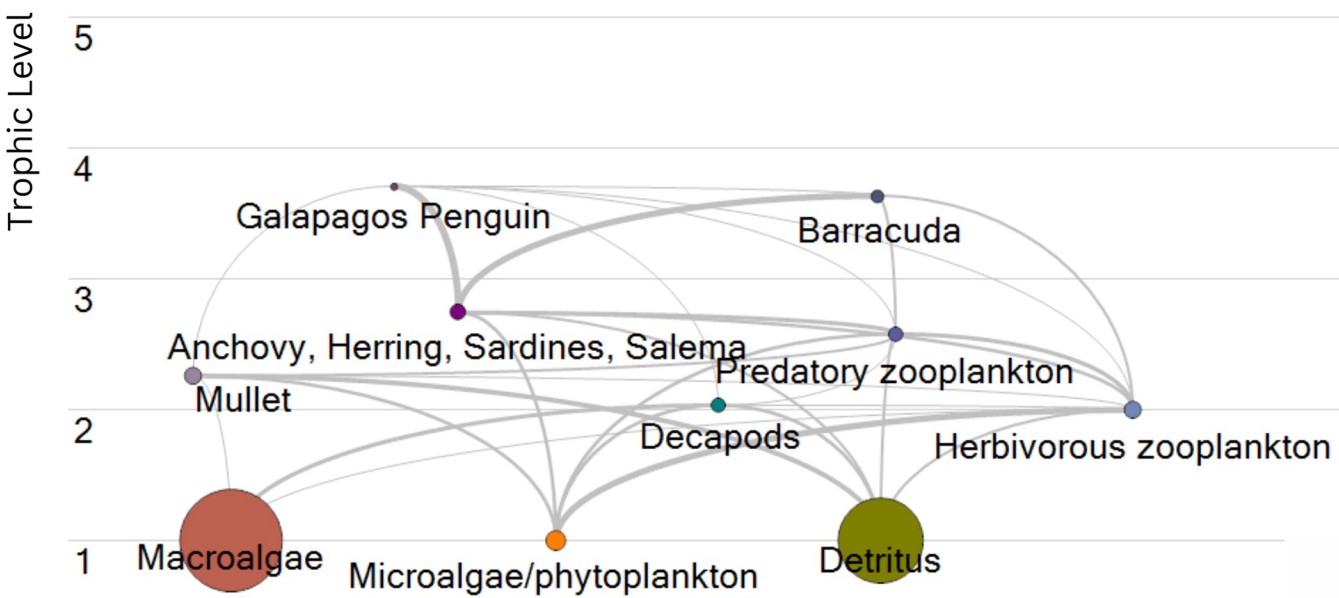

**Fig 1. Galápagos penguin EwE food web model energy flow diagram.** Circles represent the amount of biomass and labels indicate the respective functional group. Thicker lines between functional groups reflect higher energy flows from one functional group to another, and trophic levels are indicated on the y-axis.

anthropogenic particle mean concentrations for zooplankton, anchovy, and mullet were $7.38 \times 10^{-7}$, $7.69 \times 10^{-10}$, and $1.48 \times 10^{-8}$ g/kg, respectively [79].

## 3. Results

### 3.1 Galápagos penguin food web model in Ecopath

A successfully balanced GP model was constructed with a total of ten functional groups of varying trophic levels (Fig 1; Table 5). Analysis of the energy flow diagram revealed the highest energy flows from microalgae/phytoplankton to herbivorous zooplankton, as well as from anchovy, herring, sardines, and salema to barracuda and to the Galápagos penguin. A high biomass of macroalgae and detritus was necessary to maintain the balance of the model, and matched output of the BCE model [54]. Detritus, microalgae/phytoplankton, and macroalgae made up trophic level 1, followed by herbivorous zooplankton, mullet, predatory zooplankton, and anchovy, herring, sardines, salema as trophic level 2, in order of lowest to highest. Predatory zooplankton had a higher trophic level than mullets, which is expected given mullet's preference for detritus compared to predatory zooplankton consuming other zooplankton. Likewise, anchovy, herring, sardines, and salema had a similar trophic level compared to predatory zooplankton, which is not surprising given similar food preferences, namely other zooplankton. The Galápagos penguin and barracuda were the high trophic level species included in the model. Ecotrophic efficiencies (EE) were comparable to Ruiz & Wolff [54] and ranged from 0 to 0.68 where the highest EE resulted in predatory zooplankton, herbivorous zooplankton, and microalgae/phytoplankton.

### 3.2 Microplastic bioaccumulation and biomagnification via Ecotracer

**3.2.1 Galápagos penguin (GP) food web model.** Utilizing the baseline Ecotracer scenario for the GP food web model, output data revealed a rapid increase in contaminant concentration (g) per biomass (kg) until around year 5, after which it shifted to a more gradual increase

**Table 5. Parameter output from the Galápagos penguin EwE food web model.**

| | Group name | Trophic level | Habitat area (proportion) | Biomass in habitat area (t/km$^2$) | Biomass (t/km$^2$) | Production / biomass (/year) | Consumption / biomass (/year) | Ecotrophic Efficiency | Production / consumption (/year) |
|---|---|---|---|---|---|---|---|---|---|
| 1 | Galápagos Penguin | 3.710 | 1 | 0.0125 | 0.0125 | 0.067 | 60.30 | 0.000 | 0.001 |
| 2 | Barracuda | 3.636 | 1 | 13.06 | 13.06 | 0.063 | 3.9 | 0.046 | 0.016 |
| 3 | Mullet | 2.260 | 1 | 22.6 | 22.6 | 2.8 | 10.9 | 0.0006 | 0.257 |
| 4 | Anchovy, Herring, Sardines, Salema | 2.743 | 1 | 19 | 19 | 4.6 | 15 | 0.386 | 0.307 |
| 5 | Decapods | 2.035 | 1 | 14.5 | 14.5 | 0.687 | 11.95 | 0.004 | 0.058 |
| 6 | Predatory zooplankton | 2.578 | 1 | 15 | 15 | 45 | 99.1 | 0.593 | 0.454 |
| 7 | Herbivorous zooplankton | 2 | 1 | 22 | 22 | 36 | 200 | 0.566 | 0.18 |
| 8 | Macroalgae | 1 | 1 | 800.5 | 800.5 | 15.7 | | 0.023 | |
| 9 | Microalgae/ phytoplankton | 1 | 1 | 31.2 | 31.2 | 146.3 | | 0.675 | |
| 10 | Detritus | 1 | 1 | 500 | 500 | | | 0.066 | |

Parameter output captures a balanced, static representation of the food web at the moment in time with respective biomass, production / biomass (P/B), consumption / biomass (Q/B), Ecotrophic Efficiency (EE), and production / consumption (P/Q) metrics.

and eventually plateaued, reaching steady state around year 60 (Fig 2A). The microplastic concentration in the Galápagos Penguin plateaued at $7.6 \times 10^{-5}$ g/kg, while the microplastic concentration in their primary prey item, planktivorous fish, reached a steady state at $4.93 \times 10^{-8}$ g/kg. The Galápagos penguin was found to have the highest level of microplastics per biomass,

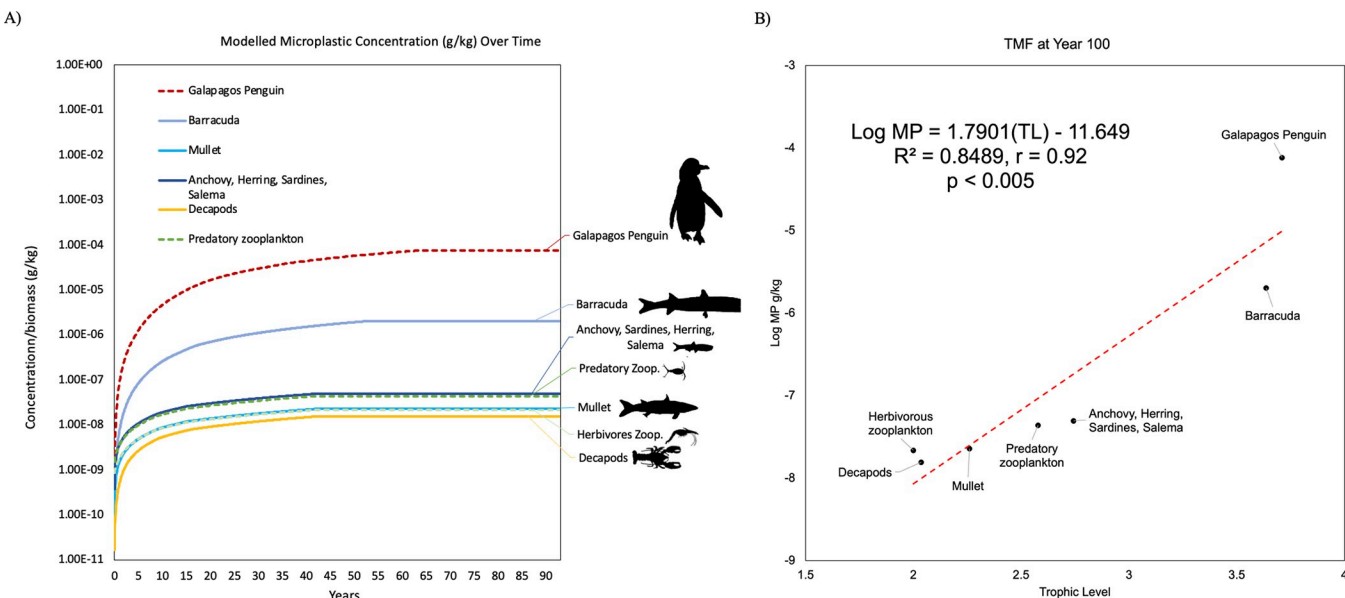

**Fig 2. Galápagos penguin model microplastic concentration over time and trophic level.** (A) Bioaccumulation simulations using the EwE Ecotracer routine, showing the projections of microplastics (MPs) bioaccumulation in the Galápagos penguins food web model under the baseline scenario with seawater initial concentrations = 0.00411 t/km$^2$. The simulations for the bioaccumulation include the elimination rates based on the literature. For zooplankton, as the key trophic level for the initial uptake of microplastics, the ingestion rate of microplastics and anthropogenic particles by zooplankton (i.e., $7.38 \times 10^{-7}$ g/kg/day) was used, based on [79]. (B) Linear regressions showing the significant relationship between predicted concentrations of microplastics (log-transformed data) and trophic levels in the GP model at year 100. The antilog of the regression slope was used to determine TMF.

followed by barracuda, anchovy, sardine, herring, and salema, and then predatory zooplankton (Fig 2A). In contrast, species or functional groups at lower trophic levels had lower concentrations of microplastics per unit of biomass.

The baseline simulation results predicted the plausible bioaccumulation and biomagnification of microplastics in the functional groups of species within this model. The relationship between the logarithmic-transformed concentration of microplastics and the species' trophic levels shows that the TMF was significantly greater than 1 (i.e., TMF > 1; when the slope is statistically different or greater than zero [($b > 0$)] in a positive, significant linear regression). The TMF increased from 7.08 at year 1 to 61.7 at year 100 (Fig 2; S5 Table in S1 File). By year 100, there was a substantial difference between the Galápagos penguin and Barracuda (trophic level 3) compared to lower trophic level groups, indicating biomagnification in the model. The regression between the logarithmic microplastic concentration (g/kg) revealed a slope of 1.79 with a TMF = 61.7 (p < 0.05), further indicating a statistically significant relationship for biomagnification in the model (Fig 2B).

The Galápagos penguin had the highest BAF, regardless of what scenario was assessed, while the mullet had the lowest BAF and BCF (Fig 2; S6 Table in S1 File). The BMF increased with each trophic level and was high ($BMF_{TL}$ = 1239 and $BMF_{TL}$ = 1796) for the Galápagos penguin, with planktivorous fish and mullet being considered as prey, respectively, though it was substantial for all prey items from the Galápagos penguin. The $BMF_{TL}$ was the lowest for planktivorous fish ($BMF_{TL}$ = 3.06) as well as for predatory zooplankton with herbivorous zooplankton as prey ($BMF_{TL}$ = 3.05).

**3.2.2 Bolivar Channel (BCE) ecosystem model.** The BCE model with the Baseline Ecotracer scenario yielded similar results to the GP food web model in terms of overall trend, but with lower microplastic concentrations per functional group. Output data once again revealed a rapid increase in contaminant concentration (g) per biomass (kg) until around year 5, and diverging slightly from the GP model, concentrations plateaued earlier, reaching steady state around year 35 as shown in Fig 3A. Microplastic concentrations in the seabirds plateaued at around $5.9 \times 10^{-7}$ g/kg, two orders of magnitude lower than results from the GP model. Microplastic concentrations in the penguins' primary prey items, planktivorous fish, reached steady states at $6.63 \times 10^{-8}$ g/kg (Fig 3A), which is within the same order of magnitude as yielded in the GP model. The functional groups with the highest concentrations of microplastics per biomass displayed a pattern consistent with the one established in the GP model. The functional groups of interest with elevated microplastic concentrations included seabirds as the highest, followed by barracuda, predatory zooplankton, and finally, small planktivorous reef fish (Fig 3A). Unlike the GP model, predatory zooplankton had slightly higher microplastic concentrations compared to small planktivorous fish and herbivorous zooplankton had higher concentrations of microplastics than both detritivorous species, mullets, and lobsters.

Similar to the GP model simulations, BCE simulation resulted in plausible bioaccumulation and biomagnification of microplastics in the functional groups of species in the model. The relationship between the concentration of microplastics and the species' trophic levels shows that the TMF was once again significantly greater than 1 (slope > 0; Fig 3A). The TMF ranged from 6.17 at year 1 to 16.6 at year 100 (S7 Table in S1 File). At year 100, the TMF was lower than estimated by the GP model. Seabirds and barracuda displayed a similar trend, yielding higher bioaccumulation and biomagnification potentials than other functional groups of interest. Of particular interest were the projections for predatory marine mammals and sharks, having the highest biomagnification capacity of all functional groups in the model. Overall, the regression indicated a high biomagnification potential as the logarithmic microplastic concentration (g/kg) versus the trophic level revealed a statistically significant slope of 1.22, i.e., TMF = 16.7 (p < 0.001; Fig 3; S7 Table in S1 File).

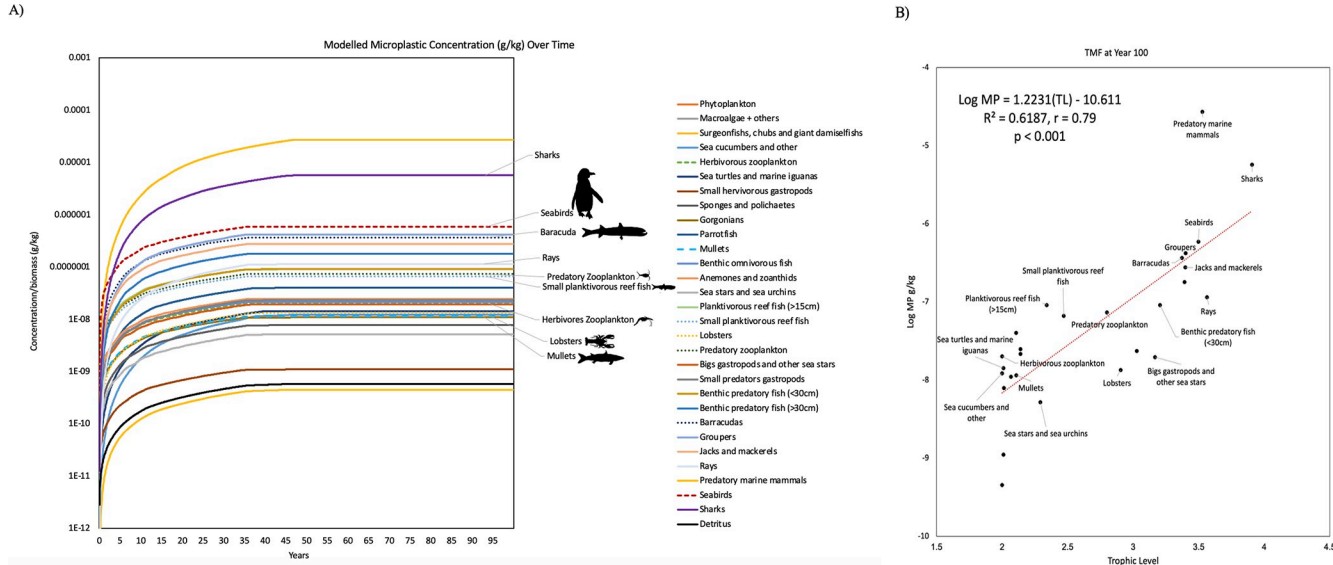

**Fig 3. BCE model microplastic concentration over time and trophic level.** (A) Bioaccumulation simulations using the EwE Ecotracer routine, showing the projections of microplastics (MPs) bioaccumulation in the BCE model under the baseline scenario where seawater initial concentrations = 0.00411 t/km². The simulations for the bioaccumulation include the elimination rates based on the literature. For zooplankton, as the key trophic level for the initial uptake of microplastics, the ingestion rate of microplastics and anthropogenic particles by zooplankton (i.e., $7.38 \times 10^{-7}$ g/kg/day) was used, based on [79]. (B) Linear regressions showing the significant relationship between predicted concentrations of microplastics (log-transformed data) and trophic levels in the BCE model at year 100. The antilog of the regression slope was used to determine TMF.

As in the GP model, seabirds had the highest BAF and BCF potential, regardless of what prey scenario was assessed, and the mullet once again had the lowest BAF and BCF (S8 Table in S1 File). The resulting $BMF_{TL}$ was different than the GP model. Foremost, the planktivorous reef fish revealed $BMF_{TL} < 1$, indicating lack of biomagnification. The Galápagos penguin revealed positive $BMF_{TL}$ across all prey items, where the highest was once again with prey item lobster. That said, $BMF_{TL}$ was substantially lower than the GP model; the highest $BMF_{TL}$ in the BCE model was a factor of 76 compared to a factor of 2308 in the GP model.

## 3.3 Model sensitivity

To assess the sensitivity of the model parameters, four different scenarios were evaluated for each model, resulting in a total of eight simulations to examine microplastic bioaccumulation and biomagnification. The simulation outcomes showed that the elimination rate was the most sensitive parameter. This was evidenced by the fact that the results from this scenario displayed the greatest variability in terms of BAF and BCF, as well as smaller ranges in $BMF_{TL}$ across both the GP model and the BCE model, as shown in Table 6.

In the 99% elimination rate scenario for both the GP and BCE models, the BAF and BCFs were substantially lower, with values that were two to four orders of magnitude lower than those estimated in the high and low abundance scenarios as well as the baseline scenarios (Table 6). The BAF and BCF for the high and low abundance simulations were comparable to the baseline scenario for the GP and BCE models.

For the GP model using the 99% elimination rate scenario, the $BMF_{TL}$ ranged from 0.02 to 49.6 depending on the predator and prey (Table 6). Conversely, the $BMF_{TL}$ had a much broader range (e.g., 3.06 to 2308) across the other three scenarios. When comparing the $BMF_{TL}$ for the Galápagos penguins and planktivorous fish, the baseline scenario resulted in the highest $BMF_{TL}$ ($BMF_{TL}$ = 1239). This value was comparable to the high and low abundance

**Table 6. Bioaccumulation, bioconcentration, and biomagnification factors in the Galápagos penguin and BCE models across scenarios.**

A) Galápagos Penguin (GP) Model

| Metric | Scenario | Predatory zooplankton with prey set to herbivorous zooplankton | Planktivorous fish with prey set to predatory zooplankton | Planktivorous fish with prey set to herbivorous zooplankton | Mullet with prey set to detritus | Galápagos penguin with prey set to planktivorous fish | Galápagos penguin with prey set to mullet | Galápagos penguin with prey set to decapods |
|---|---|---|---|---|---|---|---|---|
| BAF | Low microplastic abundance | $4.36 \times 10^{-8}$ | $4.92 \times 10^{-8}$ | $4.92 \times 10^{-8}$ | $2.27 \times 10^{-8}$ | $5.86 \times 10^{-5}$ | $5.86 \times 10^{-5}$ | $5.86 \times 10^{-5}$ |
| | High microplastic abundance | $4.36 \times 10^{-8}$ | $4.92 \times 10^{-8}$ | $4.92 \times 10^{-8}$ | $2.26 \times 10^{-8}$ | $5.89 \times 10^{-5}$ | $5.89 \times 10^{-5}$ | $5.89 \times 10^{-5}$ |
| | 99% elimination | $1.66 \times 10^{-8}$ | $2.71 \times 10^{-10}$ | $2.71 \times 10^{-10}$ | $1.15 \times 10^{-10}$ | $7.26 \times 10^{-9}$ | $7.2 \times 10^{-9}$ | $7.26 \times 10^{-9}$ |
| | Baseline | $4.36 \times 10^{-8}$ | $4.92 \times 10^{-8}$ | $4.92 \times 10^{-8}$ | $2.26 \times 10^{-8}$ | $5.90 \times 10^{-5}$ | $5.90 \times 10^{-5}$ | $5.90 \times 10^{-5}$ |
| BCF | Low microplastic abundance | $4.36 \times 10^{-8}$ | $4.92 \times 10^{-8}$ | $4.92 \times 10^{-8}$ | $2.27 \times 10^{-8}$ | $5.86 \times 10^{-5}$ | $5.86 \times 10^{-5}$ | $5.86 \times 10^{-5}$ |
| | High microplastic abundance | $4.36 \times 10^{-8}$ | $4.92 \times 10^{-8}$ | $4.92 \times 10^{-8}$ | $2.26 \times 10^{-8}$ | $5.89 \times 10^{-5}$ | $5.89 \times 10^{-5}$ | $5.89 \times 10^{-5}$ |
| | 99% elimination | $1.66 \times 10^{-8}$ | $2.71 \times 10^{-10}$ | $2.71 \times 10^{-10}$ | $1.15 \times 10^{-10}$ | $7.26 \times 10^{-9}$ | $7.26 \times 10^{-9}$ | $7.26 \times 10^{-9}$ |
| | Baseline | $4.36 \times 10^{-8}$ | $4.92 \times 10^{-8}$ | $4.92 \times 10^{-8}$ | $2.26 \times 10^{-8}$ | $5.90 \times 10^{-5}$ | $5.90 \times 10^{-5}$ | $5.90 \times 10^{-5}$ |
| $BMF_{TL}$ | Low microplastic abundance | 3.50 | 6.84 | 3.06 | 11.4 | 1232.5 | 1784 | 2300 |
| | High microplastic abundance | 3.50 | 6.85 | 3.06 | 11.3 | 1238 | 1794 | 2305 |
| | 99% elimination | 1.40 | 0.10 | 0.02 | 0.12 | 27.7 | 43.5 | 49.6 |
| | Baseline | 3.50 | 6.84 | 3.06 | 11.3 | 1239 | 1796 | 2308 |

B) Bolivar Channel Ecosystem (BCE) Model

| Metric | Scenario | Predatory zooplankton with prey set to herbivorous zooplankton | Planktivorous fish with prey set to predatory zooplankton | Planktivorous fish with prey set to herbivorous zooplankton | Mullet with prey set to detritus | Seabirds with prey set to planktivorous fish | Seabirds with prey set to mullet | Seabirds with prey set to lobster |
|---|---|---|---|---|---|---|---|---|
| BAF | Low microplastic abundance | $7.34 \times 10^{-8}$ | $6.62 \times 10^{-8}$ | $6.62 \times 10^{-8}$ | $1.14 \times 10^{-8}$ | $5.84 \times 10^{-7}$ | $5.84 \times 10^{-7}$ | $5.84 \times 10^{-7}$ |
| | High microplastic abundance | $7.34 \times 10^{-8}$ | $6.62 \times 10^{-8}$ | $6.62 \times 10^{-8}$ | $1.14 \times 10^{-8}$ | $5.85 \times 10^{-7}$ | $5.85 \times 10^{-7}$ | $5.85 \times 10^{-7}$ |
| | 99% elimination | $2.83 \times 10^{-8}$ | $3.50 \times 10^{-10}$ | $3.50 \times 10^{-10}$ | $8.45 \times 10^{-11}$ | $3.55 \times 10^{-11}$ | $3.55 \times 10^{-11}$ | $3.55 \times 10^{-11}$ |
| | Baseline | $7.34 \times 10^{-8}$ | $6.62 \times 10^{-8}$ | $6.62 \times 10^{-8}$ | $1.14 \times 10^{-8}$ | $5.85 \times 10^{-7}$ | $5.85 \times 10^{-7}$ | $5.85 \times 10^{-7}$ |
| BCF | Low microplastic abundance | $7.34 \times 10^{-8}$ | $6.62 \times 10^{-8}$ | $6.62 \times 10^{-8}$ | $1.14 \times 10^{-8}$ | $5.84 \times 10^{-7}$ | $5.84 \times 10^{-7}$ | $5.84 \times 10^{-7}$ |
| | High microplastic abundance | $7.34 \times 10^{-8}$ | $6.62 \times 10^{-8}$ | $6.62 \times 10^{-8}$ | $1.14 \times 10^{-8}$ | $5.85 \times 10^{-7}$ | $5.85 \times 10^{-7}$ | $5.85 \times 10^{-7}$ |
| | 99% elimination | $2.83 \times 10^{-8}$ | $3.50 \times 10^{-8}$ | $3.50 \times 10^{-10}$ | $8.45 \times 10^{-11}$ | $3.55 \times 10^{-11}$ | $3.55 \times 10^{-11}$ | $3.55 \times 10^{-11}$ |
| | Baseline | $7.34 \times 10^{-8}$ | $6.62 \times 10^{-8}$ | $6.62 \times 10^{-8}$ | $1.14 \times 10^{-8}$ | $5.85 \times 10^{-7}$ | $5.85 \times 10^{-7}$ | $5.85 \times 10^{-7}$ |
| $BMF_{TL}$ | Low microplastic abundance | 4.58 | -2.73 | -2.73 | 18.56 | 8.59 | 36.84 | 76.14 |
| | High microplastic abundance | 4.58 | -2.73 | -2.73 | 18.52 | 8.62 | 36.93 | 76.05 |
| | 99% elimination | 1.77 | -0.04 | -0.04 | 0.89 | 0.10 | 0.30 | 76.71 |
| | Baseline | 4.57 | -2.73 | -2.73 | 18.32 | 8.61 | 36.93 | 75.81 |

Bioaccumulation factor (BAF), bioconcentration factor (BCF), and predator-prey biomagnification factors ($BMF_{TL}$) from average microplastic concentration (g/kg) from selected predator-prey combinations in the Galápagos penguin and BCE web model, under four different scenarios including low microplastic abundance in seawater, high microplastic abundance in seawater, 99% elimination rates for all functional groups, and the baseline scenario which includes elimination rates based on available literature.

scenarios but was substantially higher than the $BMF_{TL}$ of 27.7 as estimated by the 99% elimination scenario for the same species combination.

For the BCE model, the $BMF_{TL}$ ranged from -0.04 to 76.71 in the 99% elimination rate scenario, which was more comparable to the other three scenarios yielding -2.73 to around 76 $BMF_{TL}$, depending on the predator-prey combination (Table 6). Once again, the $BMF_{TL}$ for the Galápagos penguins and planktivorous fish was much higher in the high and low abundance scenarios as well as the baseline scenarios ($BMF_{TL}$ = ~8.5) compared to the 99% elimination rate scenario ($BMF_{TL}$ = 0.10).

Simulation outcomes of the different scenarios yielded interesting results for TMF calculations, where all scenarios, except for the 99% elimination rate, predicted the plausible trophic biomagnification of microplastics in the biomass species' functional groups (Table 7; Fig 4A

**Table 7. Apparent trophic magnification factors (TMF) and regression statistics.**

| Model | Years | Scenario | Slope ($b$) | P-value | TMF (= $10^b$) | Biomagnification metric outcome |
|---|---|---|---|---|---|---|
| GP | 1 | Low Abundance | 0.85 | $p < 0.05$ | 7.07 | Potential biomagnification |
| | 25 | Low Abundance | 1.62 | $p < 0.05$ | 41.9 | Potential biomagnification |
| | 50 | Low Abundance | 1.76 | $p < 0.05$ | 57.8 | Potential biomagnification |
| | 100 | Low Abundance | 1.78 | $p < 0.05$ | 59.8 | Potential biomagnification |
| | 1 | High Abundance | 0.85 | $p < 0.05$ | 7.08 | Potential biomagnification |
| | 25 | High Abundance | 1.61 | $p < 0.05$ | 41.1 | Potential biomagnification |
| | 50 | High Abundance | 1.74 | $p < 0.05$ | 54.4 | Potential biomagnification |
| | 100 | High Abundance | 1.79 | $p < 0.05$ | 61.7 | Potential biomagnification |
| | 1 | 99% Elimination | -0.32 | $p > 0.05$ | 0.48 | Not significant/No biomagnification |
| | 25 | 99% Elimination | 0.28 | $p > 0.05$ | 1.91 | Not significant/No biomagnification |
| | 50 | 99% Elimination | 0.39 | $p > 0.05$ | 2.43 | Not significant/No biomagnification |
| | 100 | 99% Elimination | 0.41 | $p > 0.05$ | 2.57 | Not significant/No biomagnification |
| | 1 | Baseline | 0.85 | $p < 0.05$ | 7.08 | Potential biomagnification |
| | 25 | Baseline | 1.61 | $p < 0.05$ | 41.1 | Potential biomagnification |
| | 50 | Baseline | 1.74 | $p < 0.05$ | 55.4 | Potential biomagnification |
| | 100 | Baseline | 1.79 | $p < 0.05$ | 61.7 | Potential biomagnification |
| BCE | 1 | Low Abundance | 0.79 | $p < 0.05$ | 6.18 | Potential biomagnification |
| | 25 | Low Abundance | 1.19 | $p < 0.05$ | 15.6 | Potential biomagnification |
| | 50 | Low Abundance | 1.22 | $p < 0.05$ | 16.4 | Potential biomagnification |
| | 100 | Low Abundance | 1.21 | $p < 0.05$ | 16.4 | Potential biomagnification |
| | 1 | High Abundance | 0.79 | $p < 0.05$ | 6.18 | Potential biomagnification |
| | 25 | High Abundance | 1.19 | $p < 0.05$ | 15.6 | Potential biomagnification |
| | 50 | High Abundance | 1.22 | $p < 0.05$ | 16.4 | Potential biomagnification |
| | 100 | High Abundance | 1.22 | $p < 0.05$ | 16.4 | Potential biomagnification |
| | 1 | 99% Elimination | -0.72 | $p > 0.05$ | 0.19 | Not significant/No biomagnification |
| | 25 | 99% Elimination | -0.44 | $p > 0.05$ | 0.36 | Not significant/No biomagnification |
| | 50 | 99% Elimination | -0.43 | $p > 0.05$ | 0.37 | Not significant/No biomagnification |
| | 100 | 99% Elimination | -0.43 | $p > 0.05$ | 0.37 | Not significant/No biomagnification |
| | 1 | Baseline | 0.79 | $p < 0.05$ | 6.19 | Potential biomagnification |
| | 25 | Baseline | 1.20 | $p < 0.05$ | 15.7 | Potential biomagnification |
| | 50 | Baseline | 1.22 | $p < 0.05$ | 16.7 | Potential biomagnification |
| | 100 | Baseline | 1.22 | $p < 0.05$ | 16.7 | Potential biomagnification |

Apparent trophic magnification factors (TMF) and regression statistics for the linear regression models of the log of microplastic (MP) concentrations versus trophic level for the Galápagos penguin (GP) model and Bolivar Channel Ecosystem (BCE) model, for year 1, 25, 50, and 100, under four different scenarios including low microplastic abundance in seawater, high microplastic abundance in seawater, 99% elimination rates for all functional groups, and the baseline scenario which includes elimination rates based on available literature.

and 4B). For the 99% elimination rate, some slopes were negative (i.e., year 1 in the GP model and years 1 to 100 in the BCE model), indicating lack of trophic magnification, however regressions were not significant ($p > 0.05$) in the GP modelled scenario (Fig 4A) or the BCE (Fig 4B). A negative slope, if significant, would indicate trophic dilution of microplastics (i.e., a decline in microplastic concentrations as the trophic level increases, TMF < 1 [slope < 0]).

The highest TMF value was predicted at year 100 in the baseline and high abundance scenarios running in the GP model, i.e., TMF = 61.7 (Table 7). The highest TMF predicted in the BCE model was 16.7 in the high abundance and baseline scenario. Across both models under the 99% elimination rate scenario, there was very low TMF values from year one to year 100.

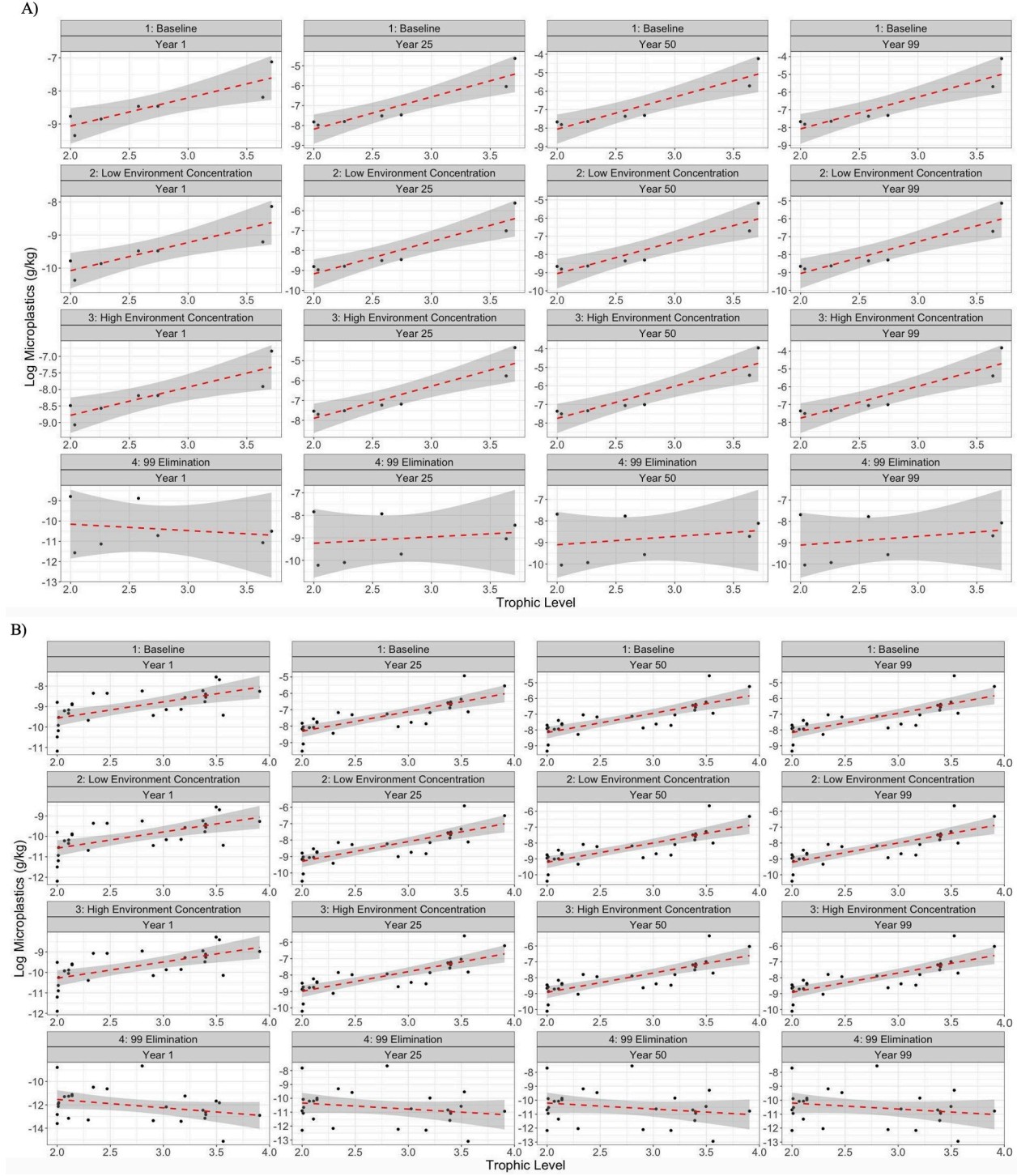

**Fig 4. Microplastic concentrations and trophic level linear regressions.** Linear regressions showing the relationship between predicted concentrations of microplastics (log-transformed data) and trophic levels for: A) the Galápagos penguin (GP) model; and, B) the Bolivar Channel Ecosystem (BCE) model, for year 1, 25, 50, and 100, under four different scenarios including low microplastic abundance in seawater, high microplastic abundance in seawater, 99% elimination rates for all functional groups, and the baseline scenario which includes elimination rates based on available literature. The antilog of the regression slope was used to determine TMF. This panel was made to visualize the overall trends, not to display the fine detail of each graph.

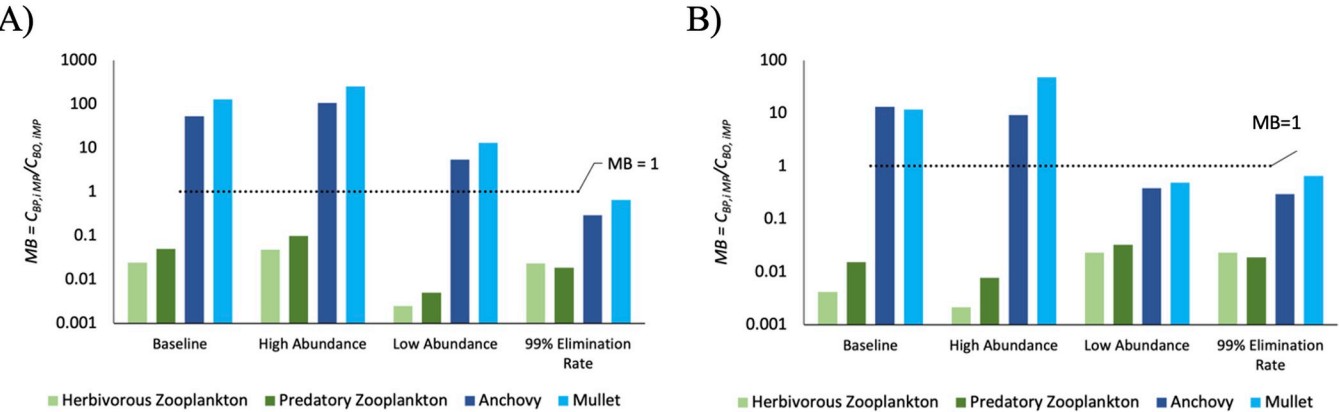

**Fig 5. Model bias.** Assessment of the model bias (MB = $C_{BP,iMP}/C_{BO,iMP}$; where $C_{BP,iMP}$ and $C_{BO,iMP}$ are the model predicted and observed microplastic concentrations, respectively, in zooplankton, anchovies, and mullets) and performance of the (A) Galápagos penguin (GP) and (B) Bolivar Channel Ecosystem (BCE) models by comparing the simulations of microplastic concentrations averaged from year 1–100. Four different scenarios are compared for model bias, including low microplastic abundance in seawater, high microplastic abundance in seawater, 99% elimination rates for all functional groups, and the baseline scenario which includes elimination rates based on available literature. The dotted line indicates MB = 1 (i.e., predicted microplastics data is equivalent to the observed microplastic concentration).

TMF ranged from 0.48 to 2.57 in the GP model, while the lowest TMF values, ranging from 0.19 to 0.37 were predicted from year one to 100 in the BCE model (Table 7). TMF rose substantially from year 1 to year 25 and then gradual increases in TMF through to year 100 were predicted for the other scenarios tested (Fig 4A and 4B).

### 3.4 Model bias

The projected concentration of MPs in the environment, and biota including zooplankton, anchovies, and mullet were compared to empirically collected data from similar abiotic (environmental) and biotic compartments. The outcomes of the *MB* ratio analysis revealed systematic under prediction (*MB* < 1) of microplastic concentrations in zooplankton, across all scenarios, with a MB ranging from 0.002 to 0.1 at low abundance scenario and high abundance scenario, respectively while using the GP model (Fig 5A and S1A, S1B Fig in S1 File). Similarly, under prediction for zooplankton occurred in the BCE model, with MB ranging from 0.002 to 0.08 in the low and baseline scenarios, respectively (Fig 5B and S1C, S1D Fig in S1 File). Conversely, the baseline, low, and high abundance (1, 2, 3) scenarios revealed over-prediction of microplastic concentrations (*MB* > 1) in fish ranging from 5.4 (low abundance) to 105.1 (high abundance) for anchovies in the GP model and 9.2 (low abundance) to 72.8 (baseline) for anchovies in the BCE model, as well as, 13.0 (low abundance) to 252.3 (high abundance) for mullets in the GP model and 11.8 (high abundance) to 65.5 (baseline) for mullets in the BCE model (Fig 5B; S1 Fig in S1 File). Results for fish are different in the 99% elimination rate scenario, namely, fish had the closest concentration values to the observed data (*MB* = ~1) (S1B and S1D Fig in S1 File). In the GP model, the 99% elimination rate scenario yielded MB of 0.29 and 0.65 and the BCE yielded MB of 0.38 and 0.48 for the anchovies and mullets, respectively.

## 4. Discussion

Ecosystem modelling simulation results in this study revealed bioaccumulation potential in all predator-prey combinations in both GP and BCE models across all scenarios. Biomagnification of microplastics was apparent in all simulations, except for the 99% elimination rate

scenario, which showed a lack of trophic magnification with insignificant negative slopes ($p > 0.05$) in both GP and BCE models. Biomagnification was highly dependent on microplastic egestion rates. As well, the microplastic concentration in zooplankton was systematically under-predicted in both GP and BCE models, whereas microplastic concentrations in fish were over-predicted, with the closest concentration values to the observed data being seen in the 99% elimination rate scenario and the baseline scenario for fish and zooplankton respectively (Fig 5 & S1 Fig in S1 File).

These results are comparable to limited existing microplastic bioaccumulation and biomagnification simulations using EwE, which is not surprising given similar methods and parameters used [93, 94]. Ma and You's [94] EwE simulations found microplastics bioaccumulate quickly in fish food webs of Baiyangdian Lake, China. Likewise, Alava et al. [93] ran Ecotracer through an extensive list of 20 marine ecosystem models available through EcoBase. The projections of microplastic concentrations in biota revealed that top predators are likely exposed to higher levels of microplastics accumulated through their diet (e.g., prey items). However, the elimination or egestion rate was likewise a key factor in determining the net bioaccumulation behaviour of microplastics. Emphasis was placed on better understanding of the role of retention times and elimination rates of microplastics in different functional groups. Likewise, microplastic bioaccumulation modelling in a cetacean food web [21] was comparable to EwE results, indicating that species-specific bioaccumulation of microplastics is likely, while biomagnification is highly dependent on species-specific elimination rates.

According to a comprehensive meta-analysis [42] evidence has been found for bioaccumulation of microplastics in marine species; however, biomagnification of these pollutants in the food web has yet to be confirmed by field data. A select number of field studies have observed trophic transfer of microplastics [16, 27, 95], but there is a disagreement as to whether microplastics are eliminated or retained in the GI tract or gut [95]. Microplastic dilution, for example, has been found in mussels and fish in the Persian Gulf [26]. To better understand this phenomenon, Miller et al. [29] explored microplastic movement through a coral reef food web and found bioconcentration evident in zooplankton, crustaceans, and fish, but no bioaccumulation or biomagnification.

Bioaccumulation and biomagnification currently represent an important debate in the field of microplastic science and uncertainty remains. While conducting research to investigate these ecotoxicological factors, it is important to consider (1) microplastics are a class of contaminants and may move differently within a food web depending on their physical and chemical characteristics [96], (2) microplastic input into the ocean is increasing year over year, and (3) evidence suggest ingestion of microplastics provides no advantage to marine organisms in a changing ocean. Thus, though microplastic biomagnification is not agreed upon, it is still important to reduce the anthropogenic emissions of microplastics to the sea.

First, it is critical to consider microplastics as a class of contaminants [96]. It is reasonable that their behaviour in the gastrointestinal tract is varied based on their characteristics (e.g., size, shape, microbial biosphere, additives, dyes), as different microplastics and associated chemicals will react differently in the gut [97, 98]. There is evidence to suggest that small microplastics and nanoplastics ($<1$ µm), not measured in this study, can pass through tissue membranes, translocate to tissues, and enter the blood stream [99–102]. Chemical additives or harmful bacteria on microplastics can adsorb within the biota [103, 104] and may or may not bioaccumulate, biomagnify, or multiply within the organism [22, 105]. Specific shapes and sizes may be more toxic [98] and prone to lower or higher GI tract retention times, influencing microplastic bioaccumulation [21].

Secondly, plastic and microplastic pollution inputs into the ocean is increasing in the Plasticene [32, 80, 106–109]. The increase in these anthropogenic particles will inevitably lead to

microplastics accumulation in ecosystems, whether it will be a marine ecosystem accessible to researchers and the public, or deep-sea ecosystems that are not commonly sampled.

Lastly, numerous studies note prominent ingestion of microplastics when organisms are exposed to plastic particles in the environment or in laboratory conditions. Microplastic ingestion has been linked with several primarily sublethal health effects [31, 34, 110, 111] and, therefore, it is plausible that ingested microplastics do not offer advantages in stressful changing oceanic conditions further harmed by overfished resources and climate change effects [112]. Instead, the presence of microplastics or nanoplastics may independently pose a significant concern and may be magnified by other environmental challenges [113].

Overall, future research should prioritize laboratory assessments of microplastic accumulation across trophic levels, with particular attention on egestion rates and GI tract retention times, physical-chemical characteristics of retained microplastics, and ecotoxicological health effects. Future modelling work should explore different interactions with microplastic at the primary consumer level. For example, microplastics can be added as a functional group in EwE and mediation applied to increase plastic consumption when phytoplankton abundance is low and vice versa [74, 93]. It would also be prudent to create model scenarios with varying rates of increase for microplastics in the specific ecosystem. Future modelling should likewise consider multiple-anthropogenic stressors, including the impact of climate change forcing and El Niño Southern Oscillation (ENSO) events in the Galápagos penguin food web when there are drastic changes on sea surface temperature (negative anomalies), disruption of primary production and pronounced food shortages when fewer fish serving as prey for penguins during El Niño are available [48, 114]. These multifactorial stressors may well affect the exposure pathways, bioaccumulation, and elimination of microplastics in marine top predators, including seabirds like the Galápagos penguins.

## 5. Conclusion

It is imperative to continue prioritizing efforts to reduce the input of microplastics into vulnerable ecosystems and food webs, such as that of the endangered Galápagos penguin. Despite ongoing research, the biomagnification of microplastics remains unclear, and additional studies are needed to fully understand this phenomenon. This study identified a knowledge gap in microplastic elimination rates, which are needed to determine biomagnification potential.

As microplastic research continues, it is key to continue efforts raising awareness and mitigating microplastic pollution entering waterways, while not losing sight of other pressing threats to the world's oceans. It is critical to adopt a balanced, holistic approach when considering oceanic threats, including plastics, but also overfishing, and climate change, in order to effectively protect and preserve iconic species like the Galápagos penguin and our precious ocean environment.

Ultimately, this trophodynamic ecosystem modelling work provided predictions and insights on the potential bioaccumulation and biomagnification risks of microplastics as a global pollutant of concern to support regional marine plastic pollution management efforts for the conservation of native and endemic species of the Galápagos Islands and the Galápagos Marine Reserve.

Please refer to the Supplementary Information for additional figures and tables, including model input values, scenario details, functional group details, linear regression data, bioaccumulation factors (BAF), bioconcentration factors (BCF), predator-prey biomagnification factors (BMF$_{TL}$), and model bias figures. All data are accessible online at SEANOE (https://doi.org/10.17882/97201) [115].

## Supporting information

**S1 File. Supplementary information includes model input values, scenario details, functional group details, linear regression data, bioaccumulation factors (BAF), bioconcentration factors (BCF), predator-prey biomagnification factors ($BMF_{TL}$), and model bias figures.**
(DOCX)

## Author Contributions

**Conceptualization:** Karly McMullen, Evgeny A. Pakhomov, Juan José Alava.

**Data curation:** Karly McMullen.

**Formal analysis:** Karly McMullen, Juan José Alava.

**Funding acquisition:** Evgeny A. Pakhomov.

**Investigation:** Karly McMullen.

**Methodology:** Karly McMullen, Juan José Alava.

**Project administration:** Karly McMullen.

**Resources:** Evgeny A. Pakhomov, Juan José Alava.

**Software:** Karly McMullen.

**Supervision:** Félix Hernán Vargas, Paola Calle, Evgeny A. Pakhomov, Juan José Alava.

**Visualization:** Karly McMullen.

**Writing – original draft:** Karly McMullen.

**Writing – review & editing:** Karly McMullen, Félix Hernán Vargas, Paola Calle, Omar Alavarado-Cadena, Evgeny A. Pakhomov, Juan José Alava.

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
