## [Decision Letter · Decision Letter 0]

9 Oct 2023

PONE-D-23-28066Modelling microplastic bioaccumulation and biomagnification potential in the Galápagos penguin ecosystem using Ecopath and Ecosim (EwE) with EcotracerPLOS ONE

Dear Dr. McMullen,

Thank you for submitting your manuscript to PLOS ONE. After careful consideration, we feel that it has merit but does not fully meet PLOS ONE’s publication criteria as it currently stands. Therefore, we invite you to submit a revised version of the manuscript that addresses the points raised during the review process.

We look forward to receiving your revised manuscript.

Kind regards,

Arumugam Sundaramanickam, PhD

Academic Editor

PLOS ONE

[A special thank you to V. Christensen and S. De la Puente Jeri for the FISH 501, an ecosystem modelling course carried out in 2021, which included phenomenal instruction and support. Authors thank and acknowledge the creators of the Bolivar Channel EwE Model, D. J. Ruiz and M. Wolff. A special thank you to D. J. Ruiz, who provided access and support throughout. Access to D. J. Ruiz and M. Wolff’s model made this research more robust. Thank you for your data transparency. Thank you to B. Hunt for offering constructive comments and feedback. K. McMullen and J. J. Alava thank the Nippon Foundation for providing funding to support fieldwork and research in Ecuador via the Nippon Foundation-Marine Litter Project at the Institute for the Oceans and Fisheries, University of British Columbia. Dr. Wilf Swartz proactively helped to manage funding and financial resources allocation and administration to support the Nippon Foundation-Marine Litter Project. This research was partially supported by the NSERC Discovery Grant RGPIN-2014-05107 to E.A.P. We humbly express our gratitude to the local people of the Galápagos Islands and the unique endemic and native species of the archipelago, who continue to inspire our desire for conservation.]

 [This research was partially funded by the Nippon Foundation via the Nippon Foundation-Marine Litter Project at the Institute for the Oceans and Fisheries, University of British Columbia, awarded to J. J. Alava. https://oceannexus.uw.edu/about/

This research was partially supported by the NSERC Discovery Grant RGPIN-2014-05107 to E. A. Pakhomov. https://www.nserc-crsng.gc.ca/professors-professeurs/grants-subs/dgigp-psigp_eng.asp

The funders had no role in study design, data collection and analysis, decision to publish, or preparation of the manuscript.]

6. We note that you have indicated that data from this study are available upon request. PLOS only allows data to be available upon request if there are legal or ethical restrictions on sharing data publicly. For more information on unacceptable data access restrictions, please see http://journals.plos.org/plosone/s/data-availability#loc-unacceptable-data-access-restrictions. 

7. We noted in your submission details that a portion of your manuscript may have been presented or published elsewhere. [Data from this article is published in K. McMullen master's thesis at the University of British Columbia, Institute for Oceans and Fisheries. It has not been published elsewhere.] Please clarify whether this [conference proceeding or publication] was peer-reviewed and formally published. If this work was previously peer-reviewed and published, in the cover letter please provide the reason that this work does not constitute dual publication and should be included in the current manuscript.

8. We note that Figure 2A and 3A in your submission contain copyrighted images. All PLOS content is published under the Creative Commons Attribution License (CC BY 4.0), which means that the manuscript, images, and Supporting Information files will be freely available online, and any third party is permitted to access, download, copy, distribute, and use these materials in any way, even commercially, with proper attribution. For more information, see our copyright guidelines: http://journals.plos.org/plosone/s/licenses-and-copyright.

A. You may seek permission from the original copyright holder of Figure 2A and 3A to publish the content specifically under the CC BY 4.0 license. 

B. If you are unable to obtain permission from the original copyright holder to publish these figures under the CC BY 4.0 license or if the copyright holder’s requirements are incompatible with the CC BY 4.0 license, please either i) remove the figure or ii) supply a replacement figure that complies with the CC BY 4.0 license. Please check copyright information on all replacement figures and update the figure caption with source information. If applicable, please specify in the figure caption text when a figure is similar but not identical to the original image and is therefore for illustrative purposes only.

Reviewers' comments:

Reviewer's Responses to Questions

**Comments to the Author**

1. Is the manuscript technically sound, and do the data support the conclusions?

Reviewer #1: Yes

Reviewer #2: Yes

2. Has the statistical analysis been performed appropriately and rigorously? 

Reviewer #1: Yes

Reviewer #2: Yes

3. Have the authors made all data underlying the findings in their manuscript fully available?

Reviewer #1: Yes

Reviewer #2: Yes

4. Is the manuscript presented in an intelligible fashion and written in standard English?

Reviewer #1: Yes

Reviewer #2: Yes

5. Review Comments to the Author

Reviewer #1: The manuscript entitled “Modelling microplastic bioaccumulation and biomagnification potential in the Galápagos penguin ecosystem using Ecopath and Ecosim (EwE) with Ecotracer” is an interesting piece of work and provides a new direction to research. There are certain points to be considered by the authors. Hence it is suggest for Minor Revision.

• The term microplastics is not defined at the first instance in the introduction.

• The overall manuscript is well written.

• The methodology is clearly written, however is very lengthy. Some information may be shifted to the supplementary section.

• Results section is very lengthy and loses focus of the primary aim of the article.

• There are numerous tables, some of which could be shifted to the supplementary material.

• Some figures have been misquoted. For example, Figure 16 A, Figure 18 B. However, such figures are not provided in the manuscript.

The manuscript can be accepted for publication after these minor revisions.

Reviewer #2: No Comments required because they organized everything very clearly. This manuscript is very interesting and it will be helpful to the science community. The authors mention everything according to the aim and objective.

6. PLOS authors have the option to publish the peer review history of their article (what does this mean?). If published, this will include your full peer review and any attached files.

Reviewer #1: No

Reviewer #2: No

---

## [Author Response · Author response to Decision Letter 0]

3 Dec 2023

Regarding the reviewers' comments, we have read and addressed each point they raised. First, the definition of "microplastics" has been incorporated in the introduction paragraph. We have reduced the methodology, shifting some details to the supplementary section, but we would like to highlight that we retained portions of the Galapagos penguin model methods; we feel that because the Galapagos penguin model is novel, data inputs need to be justified and explained thoroughly in the main manuscript. This differs from the Bolivar Channel, in which the methodology is already extensively outlined in Ruiz & Wolff (54).

We have streamlined the results section to center on the primary aim of the article, but again, we wish to point out that the Galapagos penguin model is novel. Thus, results must indicate the results of the model itself and then the bioaccumulation and biomagification results using Ecotracer. This, again, is different from the Boliver Channel model, in which the general model results were previously published in Ruiz & Wolff (54). 

We have reduced the number of tables from 13 to 7, shifting the excluded tables to supplementary information. We rectified discrepancies in figure labels and tables, ensuring alignment with the guidelines provided. 

We extend our gratitude to both reviewers and the editors for their time and evaluation of our manuscript.

---

## [Decision Letter · Decision Letter 1]

19 Dec 2023

Modelling microplastic bioaccumulation and biomagnification potential in the Galápagos penguin ecosystem using Ecopath and Ecosim (EwE) with Ecotracer

PONE-D-23-28066R1

Dear Dr. McMullen,

We’re pleased to inform you that your manuscript has been judged scientifically suitable for publication and will be formally accepted for publication once it meets all outstanding technical requirements.

Kind regards,

Arumugam Sundaramanickam, PhD

Academic Editor

PLOS ONE

Additional Editor Comments (optional):

Reviewers' comments:

Reviewer's Responses to Questions

**Comments to the Author**

1. If the authors have adequately addressed your comments raised in a previous round of review and you feel that this manuscript is now acceptable for publication, you may indicate that here to bypass the “Comments to the Author” section, enter your conflict of interest statement in the “Confidential to Editor” section, and submit your "Accept" recommendation.

Reviewer #1: All comments have been addressed

2. Is the manuscript technically sound, and do the data support the conclusions?

Reviewer #1: Yes

3. Has the statistical analysis been performed appropriately and rigorously? 

Reviewer #1: Yes

4. Have the authors made all data underlying the findings in their manuscript fully available?

Reviewer #1: Yes

5. Is the manuscript presented in an intelligible fashion and written in standard English?

Reviewer #1: Yes

6. Review Comments to the Author

Reviewer #1: The authors have addressed all the comments satisfactorily. The number of tables have been reduced and the figure numbers have been quoted appropriately. Hence it is recommended that the manuscript be accepted for publication.

7. PLOS authors have the option to publish the peer review history of their article (what does this mean?). If published, this will include your full peer review and any attached files.

Reviewer #1: No

---

## [Editor Report · Acceptance letter]

2 Jan 2024

PONE-D-23-28066R1 

PLOS ONE

Dear Dr. McMullen, 

I'm pleased to inform you that your manuscript has been deemed suitable for publication in PLOS ONE. Congratulations! Your manuscript is now being handed over to our production team.

Kind regards, 

on behalf of

Professor Arumugam Sundaramanickam 

Academic Editor

PLOS ONE